# Evaluation of the Relationship Between Air Pollutants and Emergency Department Admissions with Childhood Asthma

**DOI:** 10.3390/diagnostics14242778

**Published:** 2024-12-11

**Authors:** Yakup Söğütlü, Uğur Altaş, Tuğba Altıntaş, Zeynep Meva Altaş, Sevgi Akova, Mehmet Yaşar Özkars

**Affiliations:** 1Pediatric Emergency Medicine Clinic, Ümraniye Training and Research Hospital, University of Health Sciences, Istanbul 34764, Turkey; beyoglu@hotmail.com (Y.S.);; 2Department of Pediatric Allergy and Immunology, Umraniye Training and Research Hospital, Istanbul 34764, Turkey; myozkars@hotmail.com; 3Department of Health Management, Usküdar University, Istanbul 34662, Turkey; tugba.altintas@uskudar.edu.tr; 4Maltepe District Health Directorate, Istanbul 34841, Turkey; zeynep.meva@hotmail.com; 5Department of Public Health, International School of Medicine, Istanbul Medipol University, Istanbul 34810, Turkey

**Keywords:** asthma, children, emergency department, air pollutants

## Abstract

Background: This study aims to evaluate the relationship between the number of visits to a pediatric emergency department due to asthma attacks and air pollutants. Methods: In this ecological study, all pediatric patients who visited the pediatric emergency department of a tertiary hospital in Istanbul with asthma between January 2016 and December 2023 were included. The effect of air pollution on the number of patient visits was analyzed using a negative binomial regression model. Results: Based on the negative binomial model, a one-unit increase in SO_2_ leads to a 0.020-unit decrease in the logarithm of the number of patient visits (*p* < 0.05). A one-unit increase in NO leads to a 0.040-unit increase in the logarithm of the number of patient visits (*p* < 0.05). According to factor analysis, as the levels of NO, NOx, PM2.5, NO_2_, and PM10 in the air increase, the number of patient visits also increases; however, as the level of SO_2_ increases, the number of patient visits decreases. Conclusions: Families should be informed about environmental exposures for disease management of children with asthma. The confounding factors may also play a role in SO_2_ level and the decrease in admissions due to asthma. Further studies are needed in this regard.

## 1. Introduction

Asthma is one of the most common non-communicable diseases affecting adults and children [1]. Asthma, which is a chronic inflammatory condition in the respiratory tract, may lead to symptoms such as wheezing, shortness of breath, cough, and chest pain [2]. Asthma may lead to a decrease in quality of life, sleep disorders, and psychosocial problems [3,4].

The prevalence of asthma is increasing worldwide [5]. Asthma is the most common chronic disease of childhood [1]. The Centers for Disease Control and Prevention (CDC) reported that 1 in 12 children in the United States of America (USA) has asthma [6]. An asthma attack is a sudden worsening of asthma symptoms. Again, according to CDC data, half of the children diagnosed with asthma experience an asthma attack at least once a year [6]. Patients experiencing acute asthma exacerbation may require urgent medical intervention, depending on the severity of their symptoms [7]. Factors such as asthma severity, asthma treatments used, and recurrent hospital admissions due to asthma have been reported to be associated with hospital admissions due to asthma attacks [8]. In addition, environmental factors such as seasonal causes and air pollutant amounts may increase the number of admissions to the emergency department due to asthma attacks [9,10].

Air pollution remains one of the most significant environmental threats to children’s health. Air pollution is the second most significant risk factor for non-communicable diseases and plays a crucial role in protecting public health [11]. Exposure to elevated levels of air pollution can lead to a range of adverse health effects [12]. Air pollutants are classified into gaseous pollutants and particulate matter (PM). The primary gaseous pollutants consist of inorganic components such as nitrogen dioxide (NO_2_), sulfur dioxide (SO_2_), ozone (O_3_), carbon monoxide (CO), and carbon dioxide (CO_2_) [13]. The relationship between air pollution and asthma development is known [13]. A study in the literature reported a stronger association between asthma morbidity and air pollution in children compared to adolescents and adults [14].

When inadequately treated, asthma carries even a significant risk of mortality [15]. Therefore, understanding the factors that may be associated with asthma attacks is of critical importance in asthma management. This study aims to evaluate the relationship between the number of visits to a tertiary hospital’s pediatric emergency department due to asthma attacks and air pollutants.

## 2. Materials and Methods

### 2.1. Study Design and Population

In our ecological type of study, all pediatric patients who visited the pediatric emergency department of a tertiary hospital in Istanbul with asthma between January 2016 and December 2023 were included. The time of visit, age, and gender of the patients were evaluated retrospectively.

The International Classification of Diseases, 10th Revision codes (ICD-10), J45.00, and its subcodes were accepted for the inclusion of asthma patients. All patients with related ICD codes were included without exclusion criteria. The diagnosis of asthma in children visiting the emergency department was made by the physician based on the physical examination and clinical history of the patient. Acute asthma attack was treated in the emergency department. In the subsequent phase, children were referred to the pediatric outpatient clinics for the follow-up and management of asthma disease.

Data on air pollution (PM10, PM2.5, SO_2_, NO, NO_2_, NOx) for Ümraniye, the district where the study was conducted, were obtained from the official website of the Ministry of Environment, Urbanization, and Climate Change [16]. The air pollution data were obtained from an air pollution monitoring station located in the same neighborhood as the hospital where the study was conducted. The measurements are not individual-based but area-based, representing data from the monitoring region [16].

### 2.2. Statistical Analysis

SPSS (Statistical Package for Social Sciences) for Windows 25.0 program was used for statistical analysis and data recording. Descriptive data were presented with mean, standard deviation, minimum, and maximum values. The dependent variable was the number of patient visits, and the independent variables were PM10, PM2.5, SO_2_, NO_2_, NOx, and NO (air pollution variables). The effect of air pollution on the number of patient visits was analyzed using a negative binomial regression model. Since the dependent variable, the number of patient visits, is a discrete variable, Poisson regression was initially considered as the base model. However, Poisson distribution assumes that the mean and variance are equal. In the preliminary analyses, it was observed that the variance in the dataset was larger than the mean, indicating the presence of an overdispersion problem. In cases of overdispersion, the Poisson model becomes insufficient and can produce biased estimates.

To address this issue, negative binomial regression, which accounts for overdispersion, was employed. The negative binomial model is a generalized version of Poisson regression, appropriate for situations where the variance exceeds the mean, making the model more flexible and better suited to the data.

To assess the goodness of fit of the model, the Omnibus Test, Deviance/sd, and Pearson χ^2^/sd values were used. The lack of significant effects for some independent variables in the initial model suggested the presence of multicollinearity, and the relationship between the numerical variables was evaluated using Pearson correlation analysis. Variables with significant relationships were reduced to fewer dimensions through factor analysis, and the negative binomial regression model was re-analyzed using the factors representing the independent variables. A *p* value of <0.05 was considered to be the statistical significance level.

### 2.3. Ethics

The ethics committee approval for the study was granted by the ethics committee of the relevant institution (date: 13 June 2024, number: 170). Since data were obtained retrospectively from the system records without communicating with the patients, informed consent form was not obtained.

## 3. Results

The dataset of the study consists of the number of incoming patients between 2016 and 2023. The data were obtained monthly. The data for air pollution variables were collected daily, but since the number of patient admissions is monthly, the air pollution variables were obtained by calculating their monthly averages. The dataset of the study consists of the number of incoming patients and air pollution variables between 2016 and 2023. The total number of patients who came between the relevant years was 84,893, 59% of whom were male and 41% were female. When the distribution by age is analyzed, 71.8% of the patients were between the ages of 0 and 5 years, 19.9% were between the ages of 6 and 11 years, and 8.3% were between the ages of 12 and 17 years.

The dependent variable was the number of incoming patients, and the independent variables were PM10, PM2.5, SO_2_, NO_2_, NOx, and NO (air pollution variables), and the effect of air pollution on the number of incoming patients was analyzed. Since the dependent variable is discrete (counting number) and does not fit the Poisson distribution (variance > mean), there was overspread, and the negative binomial regression model was preferred. Descriptive statistics of the dependent and independent variables are presented in Table 1.

The descriptive analysis revealed significant variability in both the number of patients and air pollution levels. The monthly number of patients admitted ranged from 39 to 4369, with a mean of 926.30 and a standard deviation of 885.66, indicating substantial fluctuations. Among the air pollution variables, PM10 concentrations ranged from 16.34 to 70.62 µg/m^3^ (mean = 39.99, SD = 11.69), while PM2.5 levels were between 6.66 and 37.15 µg/m^3^ (mean = 18.72, SD = 6.48). SO_2_ exhibited the highest variability, with concentrations ranging from 2.24 to 90.82 µg/m^3^ (mean = 10.59, SD = 15.99). Similarly, NO_2_ ranged from 17.27 to 110.51 µg/m^3^ (mean = 61.29, SD = 20.74), NOx from 58.21 to 333.38 µg/m^3^ (mean = 163.00, SD = 72.90), and NO from 16.61 to 160.04 µg/m^3^ (mean = 66.81, SD = 35.55).

In terms of model fit statistics, both the Deviance and Pearson χ^2^ values divided by their degrees of freedom (Deviance/sd and Pearson χ^2^/sd) were well below 1. This indicates that the model fits the data well, there is no overdispersion issue, and the negative binomial regression model can be appropriately used to explain the dependent variable, which is the number of patient visits. According to the results of the Omnibus Test conducted to assess the impact of air pollution variables (PM10, PM2.5, SO_2_, NO_2_, NOx, NO) on the number of patient visits and to test the goodness of fit of the model, the model’s goodness of fit statistic was found to be significant (χ^2^ = 52.560, df = 6, *p* < 0.001). Based on the negative binomial model, a one-unit increase in SO_2_ leads to a 0.020-unit decrease in the logarithm of the number of patient visits. This effect is significant (*p* < 0.05), and the variable has a negative impact. A one-unit increase in NO leads to a 0.040-unit increase in the logarithm of the number of patient visits. This effect is significant (*p* < 0.05), and NO has a positive effect on the number of patient visits. As the NO level increases, the number of patient visits is predicted to increase. While SO_2_ and NO have statistically significant effects in predicting the number of patient visits, the effects of PM10, PM2.5, NO_2_, and NOx are not significant (Table 2).

For Deviance, the value is 39.440, df: 82, value/df: 0.481; for Pearson χ^2^, the value is 36.811, df: 82, value/df: 0.449.

The equation for the negative binomial regression model is as follows:log(Patient number) = 5.211 + 0.002PM10 + 0.031PM2.5 − 0.020SO_2_ + 0.013NO_2_ − 0.016NOx + 0.040NO.

There is a positive and significant relationship between PM10 and PM2.5, NO_2_, NOx, and NO. Similarly, there is a positive and significant relationship between PM2.5 and NO_2_, NOx, and NO. In contrast, SO_2_ has a negative and significant relationship with NO_2_, NOx, and NO. Furthermore, there is a positive and significant relationship between NO_2_ and NOx, as well as between NO_2_ and NO. The relationship between NOx and NO is also positive and significant. The correlation coefficients for the independent variables are presented in Table 3.

To better understand the impact of air pollution on the number of patient visits, the independent variables that are correlated with each other were combined using factor analysis. According to the results of the factor analysis, the dataset was deemed suitable for this method. The Kaiser–Meyer–Olkin (KMO) measure was 0.605, indicating that the sample size is adequate for factor analysis, as values above 0.60 are considered acceptable. The chi-square value obtained from Bartlett’s test was 505.606 (*p* < 0.05), confirming that the correlations between the variables are significant and that factor analysis is appropriate.

The results of the factor analysis indicated that two components explained 78.4% of the total variance. As a result of the analysis, the variables were grouped into two factors, and Table 4 shows which variables belong to each factor. The first factor has high loadings on NO (0.939), NOx (0.990), PM2.5 (0.793), NO_2_ (0.785), and PM10 (0.678), while the second factor shows a high loading on SO_2_ (0.851). These findings suggest that the air pollution variables can be reduced into two primary factors.

Instead of including all independent variables individually in the model, the two factors representing them were included as independent variables, and the negative binomial regression analysis was conducted again. In terms of the model’s goodness of fit statistics, both the Deviance and Pearson χ^2^ values divided by their degrees of freedom (Deviance/df and Pearson χ^2^/df) were well below 1. This indicates that the model fits the data well, there is no overdispersion issue, and the negative binomial regression model can be appropriately used to explain the dependent variable, which is the number of patient visits.

According to the results of the Omnibus Test, conducted to evaluate the impact of the two factors formed by the air pollution variables on the number of patient visits and to assess the model’s goodness of fit, the goodness of fit statistic was found to be significant (χ^2^ = 41.716, df = 2, *p* < 0.001). These findings indicate that both factors are significant in the model. Factor 1 (NO, NOx, PM2.5, NO_2_, PM10) has a positive effect, while Factor 2 (SO_2_) has a negative effect. In other words, as the levels of NO, NOx, PM2.5, NO_2_, and PM10 in the air increase, the number of patient visits also increases; however, as the level of SO_2_ increases, the number of patient visits decreases. The parameter estimates of the model are presented in Table 5.

For Deviance, the value is 50.283, df: 86, value/df: 0.585; for Pearson χ^2^, the value is 53.308, df: 86, value/df: 0.620.

## 4. Discussion

The control of factors that may trigger diseases and increase morbidity, in addition to appropriate treatment, is of utmost importance in disease management. Environmental exposures are one of these critical factors. Today, air pollution is a significant public health concern, associated with increased morbidity and mortality in many diseases. Assessing the impact of air pollution on children with asthma is vital for effective disease control.

In this study, which examines the relationship between the number of emergency visits by children with asthma and changes in air pollutants, 59% of the children who presented to the emergency department for asthma between 2016 and 2023 were male, while 41% were female. The literature indicates that the risk of asthma is lower in girls under the age of 10 but equalizes during puberty [17]. The slightly higher male ratio in our study is consistent with these findings. When analyzing the age distribution of the study group, 71.8% of the children were between 0 and 5 years old, while the lowest number of visits was observed in the 12–17 years age group (8.3%). Similarly, studies in the literature also report that the majority of children presenting to the emergency department for asthma are under 5 years old [9,18]. Another study also reported that the risk of asthma attacks is higher in children under 5 years of age [19]. Therefore, the risk of asthma attacks should be evaluated in detail, especially in children diagnosed with asthma under the age of 5 years, and the family should be informed about asthma attacks. Informing the family about the management of possible attacks will also help prevent unnecessary visits to the emergency room due to excessive parental anxiety, especially in this age group.

In this study, the relationship between air pollutants and asthma visits to emergency departments was evaluated. An increase in NO, NOx, PM2.5, NO_2_, and PM10 values in the air was observed with an increase in the number of asthma-related emergency room visits. In a study in the literature, it was reported that PM may cause asthma exacerbations in previously susceptible individuals by interacting with allergens in the air [20]. In different studies, PM2.5 and PM10 values were found to be associated with asthma attacks [21,22]. In another study, PM10 was identified as a significant risk factor, contributing to a 2% rise in asthma-related emergency visits among children [23]. NO_2_ is another respiratory irritant that can penetrate deep into the lungs, leading to respiratory conditions such as coughing, wheezing, shortness of breath, bronchospasm, and even pulmonary edema when inhaled at high concentrations [24]. According to the results of our study and the literature, we observe that air pollution worsens asthma; therefore, combating air pollution has an important place in asthma management. We suggest that multidisciplinary approaches should be adopted in order to prevent the increasing disease burden of both air pollution and asthma.

A systematic review demonstrated a strong association between SO_2_ exposure and an increase in moderate to severe asthma exacerbations in children [25]. In another study, long-term SO_2_ exposure was similarly associated with the development of asthma and atopy. In the same study, long-term SO_2_ exposure was also reported to be associated with decreased FEV1/FVC ratios, more prominently in children with asthma [26]. Similarly, in a study conducted in China, NO_2_, SO_2_, and CO levels in the air showed a positive association with the number of asthma-related outpatient and emergency room visits in children [27]. In our study, a decrease in the number of emergency room visits of asthma patients was observed with an increase in SO_2_ levels. It is thought-provoking that our study found a contrasting relationship with the results in the literature. There may be several reasons for this situation. According to the correlation analysis in our study, as the SO_2_ value in the air increased, other air pollutants except PM10 decreased significantly. Also, in the regression analysis, the factor loading of SO_2_ was lower than the factor loading of other air pollutants. The decrease in asthma admissions with an increase in SO_2_ in the air is due to the increase in other air pollutants with higher factor loadings. In other words, the decrease in other airborne pollutants (NO, NOx, PM2.5, NO_2_) may have been greater than the increase in SO_2_, leading to a decrease in asthma admissions. SO_2_ is highly soluble in water and can cause significant damage to the upper respiratory tract and skin. It is also possible that our sample did not experience asthma exacerbation, but rather other harms of SO_2_, such as upper respiratory tract or skin irritation [24]. In addition, the human body’s reactions to air pollutant exposure are influenced by several factors, including the type of pollutants, the duration and intensity of exposure, atmospheric conditions, and individual characteristics [24]. We recommend further studies to measure SO_2_ levels in the air in different regions and to evaluate its relationship with emergency admissions due to asthma.

### Limitations and Strengths

Among the strengths of this study is that it has a large sample. Examining the effect of changes in air pollutants on emergency applications is another strength of the study since there is no current literature data in this field in our province. Thus, the study may provide valuable perspectives for clinicians and decision-makers working in the field of environmental health.

This study also has some limitations. In this study, the relationship between air pollutants and asthma visits to the emergency department was evaluated at the societal level. Individual exposure levels and other individual factors (genetic predisposition, indoor air quality, and smoking status) were not taken into account. In this case, the possibility of ecological bias, which can be seen in ecological studies, should be taken into account when generalizing the findings. Another limitation of the study is that although it provides a large sample size, the study was conducted in a single tertiary healthcare institution. Asthma admissions to a tertiary care hospital may have been more severe cases, in which case the change in milder asthma attacks with increasing air pollutants may not have been well assessed. Further multicenter studies will provide more generalizable findings. Another limitation of our study is the fact that some patients seek care at hospitals outside their area of residence, while we assume that most emergency department admissions occur at the nearest hospital. Nevertheless, since the visits were emergency department admissions, it is likely that individuals sought care at the hospital nearest to their place of residence. However, this limitation should also be taken into account when interpreting the results. The lack of consideration for lag times between exposure to air pollutants and the need for hospital admission is another limitation of our study.

## 5. Conclusions

In this study examining the relationship between air pollutants and the number of asthma-related emergency room visits, an increase in NO, NOx, PM2.5, NO_2_, and PM10 values was found to be associated with an increase in asthma-related emergency room visits. Families should be informed about environmental exposures in the treatment and disease management of children with asthma. In addition, our study findings may lead to multidisciplinary interventions to prevent air pollution.

In this study, a decrease in the number of admissions of asthma patients was observed with an increase in SO_2_ level. The lower factor loading of SO_2_ in the analysis suggests that confounding factors also play a role between SO_2_ level and the decrease in admissions due to asthma. In order to better understand this issue, further studies should be planned to examine genetic factors, smoking, and air pollutant exposures at home and school where children spend a lot of time, asthma medication use, and asthma severity.

## Figures and Tables

**Table 1 diagnostics-14-02778-t001:** Descriptive statistics.

		Minimum	Maximum	Mean	Standard Deviation
Dependent variable	Number of patients	39.00	4369.00	926.30	885.66334
Independent variables	PM10	16.34	70.62	39.99	11.69
	PM2.5	6.66	37.15	18.72	6.48
	SO_2_	2.24	90.82	10.59	15.99
	NO_2_	17.27	110.51	61.29	20.74
	NOx	58.21	333.38	163.00	72.90
	NO	16.61	160.04	66.81	35.55

**Table 2 diagnostics-14-02778-t002:** Parameter estimates.

Parameters	B	Standard Error	Wald χ^2^	sd	*p*
Constant	5.211	0.4707	122.574	1	<0.001
PM10	0.002	0.0112	0.032	1	0.857
PM2.5	0.031	0.0245	1.613	1	0.204
SO_2_	−0.020	0.0063	9.757	1	0.002
NO_2_	0.013	0.0112	1.429	1	0.232
NOx	−0.016	0.0096	2.833	1	0.092
NO	0.040	0.0172	5.486	1	0.019

**Table 3 diagnostics-14-02778-t003:** Correlations between independent variables.

		PM10	PM2.5	SO_2_	NO_2_	NOx	NO
PM10	Pearson correlation	1	0.543	0.107	0.292	0.475	0.516
	*p*		<0.001	0.316	0.004	<0.001	<0.001
PM2.5	Pearson correlation	0.543	1	−0.065	0.495	0.598	0.660
	*p*	<0.001		0.542	<0.001	<0.001	<0.001
SO_2_	Pearson correlation	0.107	−0.065	1	−0.279	−0.243	−0.205
	*p*	0.316	0.542		0.008	0.022	0.054
NO_2_	Pearson correlation	0.292	0.495	−0.279	1	0.848	0.761
	*p*	0.004	<0.001	0.008		<0.001	<0.001
NOx	Pearson correlation	0.475	0.598	−0.243	0.848	1	0.974
	*p*	<0.001	<0.001	0.022	<0.001		<0.001
NO	Pearson correlation	0.516	0.660	−0.205	0.761	0.974	1
	*p*	<0.001	<0.001	0.054	<0.001	<0.001	

**Table 4 diagnostics-14-02778-t004:** Variables and factor loadings.

Factor	Variables	Factor Loading	
1	NO	0.939	−0.159
	NOx	0.930	−0.247
	PM2.5	0.793	0.165
	NO_2_	0.785	−0.394
	PM10	0.678	0.488
2	SO_2_	−0.133	0.851

**Table 5 diagnostics-14-02778-t005:** Parameter estimates.

Parameters	B	Standard Error	Wald χ^2^	sd	*p*
Constant	6.597	0.1061	3866.217	1	<0.001
Factor 1 (NO, NOx, PM2.5, NO_2_, PM10)	0.637	0.1034	37.934	1	<0.001
Factor 2 (SO_2_)	−0.277	0.1322	4.381	1	0.036

## Data Availability

The data presented in this study are available on request from the corresponding author. The data are not publicly available due to ethical restrictions.

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
