# Peer review of "Evaluation of the Relationship Between Air Pollutants and Emergency Department Admissions with Childhood Asthma"

_diagnostics, 2024, doi:10.3390/diagnostics14242778_

Round 1
Reviewer 1 Report
Comments and Suggestions for Authors
I am critical about this study because of the following reasons:
- General knowledge of the authors about air pollutants and how they are monitored seems inadequate. In Lines 50-51, gases are gases and there are no particles to say their size distribution influence the definition. Exposure to ambient air pollutants do not lead to immediate need for hospital admission, unless after an exceptional episode (e.g., natural fire). There is a lag between exposure to air pollutants and potentially need for susceptible individuals to get admitted to a hospital. These are all neglected in this study.
- Rationale for selecting statistical analyses is missing.
- Literature review is incomplete, and they needed to cite many other works in this line of study. Line 57 and 35, in particular, need citation.
- Measure for hospital admission is unclear. What type of visit is being monitored? Are they in-patient or out-patient admissions? How do we know whether incoming patients are coming from the area at which those air pollutants were measured? Patients usually go to the health care facility their insurance covers, or they know the doctor. Geographical relevance between air pollutants and those admitted at a hospital is highly under question.
Author Response
Point 1: General knowledge of the authors about air pollutants and how they are monitored seems inadequate. In Lines 50-51, gases are gases and there are no particles to say their size distribution influence the definition. Exposure to ambient air pollutants do not lead to immediate need for hospital admission, unless after an exceptional episode (e.g., natural fire). There is a lag between exposure to air pollutants and potentially need for susceptible individuals to get admitted to a hospital. These are all neglected in this study.
Answer 1: Dear Reviewer, thank you for your valuable comments. We changed the sentence in lines 50-51 as following: “Air pollutants are classified into gaseous pollutants and particulate matter (PM).”
While our study did not explicitly analyze lag times, we believe that air pollution levels in the short preceding period are likely to have been similar to those observed on the day of hospital admissions, given the general stability of air quality trends in the study region. However, we acknowledge that the lack of consideration for lag times is a limitation of our study. This has been noted in the manuscript (limitations part). We will address this issue in future research to better capture the temporal relationship between exposure and hospital visits.
Point 2: Rationale for selecting statistical analyses is missing.
Answer 2: We have added this information to results part: “Since the dependent variable is discrete (counting number) and does not fit the Poisson distribution (variance>mean), there was over-spread and negative binomial regres-sion model was preferred instead of Poisson loglinear model.”
Point 3: Literature review is incomplete, and they needed to cite many other works in this line of study. Line 57 and 35, in particular, need citation.
Answer 3: We have added the references in needed areas as you suggested.
Point 4: Measure for hospital admission is unclear. What type of visit is being monitored? Are they in-patient or out-patient admissions? How do we know whether incoming patients are coming from the area at which those air pollutants were measured? Patients usually go to the health care facility their insurance covers, or they know the doctor. Geographical relevance between air pollutants and those admitted at a hospital is highly under question.
Answer 4: In our study, we specifically analyzed emergency department visits related to childhood asthma, focusing solely on outpatient admissions. We acknowledge that factors such as patient preference for specific doctors or healthcare facilities covered by insurance might influence admissions, and these limitations are noted in our study. Nevertheless, since the visits were emergency department admissions, it is likely that individuals sought care at the hospital nearest to their place of residence. Additionally, the air pollution monitoring station is located in the same neighborhood as the hospital where the study was conducted.
Reviewer 2 Report
Comments and Suggestions for Authors
Overall this is an interesting piece of work - designed to look again at the interactions between pollution exposure and asthma
Major points
1. Pollution data - there is no discussion on what the data is - is it the yearly averages? or the 3 monthly average - this needs to be clearly articulated - also are we looking at a lag - does this represent pollution for the period of time that you looked at
2. analysis of the pollution data
- is this modelled ? or fixed? related to the participant directly? or to the area? or the hospital?
Author Response
Point 1: Pollution data - there is no discussion on what the data is - is it the yearly averages? or the 3 monthly average - this needs to be clearly articulated - also are we looking at a lag - does this represent pollution for the period of time that you looked at
Answer 1: Dear Reviewer, thank you for your valuable comments. “The data set of the study consists of the number of incoming patients between 2016 and 2023. The data were obtained monthly. The data for air pollution variables was collected daily, but since the number of patient admissions is monthly, the air pollution variables were obtained by calculating their monthly averages.” We have added this information to the first paragraph of the results part.
Point 2: analysis of the pollution data - is this modelled ? or fixed? related to the participant directly? or to the area? or the hospital?
Answer 2: The data for air pollution variables was collected daily, but since the number of patient admissions is monthly, the air pollution variables were obtained by calculating their monthly averages. The air pollution data were obtained from an air pollution monitoring station located in the same neighborhood as the hospital where the study was conducted. The measurements are not individual-based but are area-based, representing data from the monitoring region. This information was added to methods and results part.
Reviewer 3 Report
Comments and Suggestions for Authors
I read the manuscript with great interest.
It addresses an important issue regarding the impact of air pollution on the severity of asthma symptoms in children. This study showed that an increase in the levels of NO, NOx, PM2.5, NO2, and PM10 in the air is associated with an increase in the number of visits due to asthma. The authors suggest that families should be informed of the impact of environmental pollution on the health of children with asthma.
I believe that the manuscript can be accepted in its current form.
This was supported by the following strengths:
1. large study group
2. appropriate statistical analyses in an ecological study design
3. Practical aspect: these results may be useful for clinicians and decision makers in the field of environmental health.
The study also has limitations related to the ecological study design, but the authors discussed them and indicated them in the discussion.
I believe that the manuscript is valuable, meets the requirements of scientific publication, and does not require any changes.
Author Response
Point 1: I believe that the manuscript can be accepted in its current form. This was supported by the following strengths:
- large study group
- appropriate statistical analyses in an ecological study design
- Practical aspect: these results may be useful for clinicians and decision makers in the field of environmental health.
The study also has limitations related to the ecological study design, but the authors discussed them and indicated them in the discussion.
I believe that the manuscript is valuable, meets the requirements of scientific publication, and does not require any changes
Answer 1: Dear Reviewer, Thank you for your valuable comments.
Round 2
Reviewer 1 Report
Comments and Suggestions for Authors
Authors tried to address my comments, but I still do not see this work novel and useful amply for publication. Nothing new (both in terms of the statistical methods used and results) has been presented. I do not suggest publishing this manuscript.
Reviewer 2 Report
Comments and Suggestions for Authors
My comments have been addressed - I am happy to approve